# Rapid Identification of Potential Drug Candidates from Multi-Million Compounds’ Repositories. Combination of 2D Similarity Search with 3D Ligand/Structure Based Methods and In Vitro Screening

**DOI:** 10.3390/molecules26185593

**Published:** 2021-09-15

**Authors:** Katalin Szilágyi, Beáta Flachner, István Hajdú, Mária Szaszkó, Krisztina Dobi, Zsolt Lőrincz, Sándor Cseh, György Dormán

**Affiliations:** TargetEx Ltd., Madách I. u. 31/2, 2120 Dunakeszi, Hungary; szilagyi@target-ex.com (K.S.); flachner@target-ex.com (B.F.); hajdu@target-ex.com (I.H.); szaszko@target-ex.com (M.S.); dobi@target-ex.com (K.D.); lorincz@target-ex.com (Z.L.); cseh@target-ex.com (S.C.)

**Keywords:** 2D similarity search, virtual screening, pharmacophore matching, 3D modelling, in vitro screening

## Abstract

Rapid in silico selection of target focused libraries from commercial repositories is an attractive and cost-effective approach in early drug discovery. If structures of active compounds are available, rapid 2D similarity search can be performed on multimillion compounds’ databases. This approach can be combined with physico-chemical parameter and diversity filtering, bioisosteric replacements, and fragment-based approaches for performing a first round biological screening. Our objectives were to investigate the combination of 2D similarity search with various 3D ligand and structure-based methods for hit expansion and validation, in order to increase the hit rate and novelty. In the present account, six case studies are described and the efficiency of mixing is evaluated. While sequentially combined 2D/3D similarity approach increases the hit rate significantly, sequential combination of 2D similarity with pharmacophore model or 3D docking enriched the resulting focused library with novel chemotypes. Parallel integrated approaches allowed the comparison of the various 2D and 3D methods and revealed that 2D similarity-based and 3D ligand and structure-based techniques are often complementary, and their combinations represent a powerful synergy. Finally, the lessons we learnt including the advantages and pitfalls of the described approaches are discussed.

## 1. Introduction: The Emergence of Virtual Screening

Virtual screening (VS) has become a popular technique [1] since it was expected to reduce the synthesis and biological screening cost and shorten the life cycles of the discovery phases. Over the past two decades, huge compound repositories were built exploiting historical collections as well as compound libraries. At the same time, the chemoinformatics methods have developed rapidly and the computational power increased, allowing fast or real time calculations. In addition, deeper knowledge has been developed about the small molecule–protein interactions using state-of-the-art X-ray crystallography, docking, and 3D modelling [2]. The major elements and the most important VS approaches are shown in Figure 1. The in vitro screening of the focused libraries often result in a many fold increase in the hit rate compared with random screening of commercial libraries [3,4].

The above approaches vary in computational requirements and hit diversity/novelty [5]. The various virtual screening methods are compared in Table 1.

The method of 2D ligand-based (LB) similarity searching involves the retrieval of molecules in a database that are structurally similar to reference compounds. It can be carried out on a regular computer requiring minimal setup and configuration and it is less CPU-intensive than other methods. Most of the commonly used fingerprints are calculated based on 2D structures; thus, conformations do not need to be generated [6]. Speed and applicability to any target class represent marked advantages, particularly if insufficient target structure is available but at least one known ligand exists [7]. The speed and simplicity also allows to screen large datasets.

On diverse compound activity classes, similarity searching using 2D fingerprints systematically produces a higher number of compounds than docking calculation and its computational efficiency is coupled with demonstrated effectiveness in many comparative studies [8]. In the last decade, SB VS applications tended to dominate the VS field partly due to the fact that available 3D structure information has been progressively increasing, but the success rate was rather moderate: large numbers of the docking hits were found only weakly potent [9]. Nevertheless, if ligand information is scarce, and there is sufficient structural information about the targets (X-ray with sufficient resolution), 3D docking would provide more diverse virtual hits and increased novelty than techniques that rely on existing active compounds (such as 2D similarity selection). In order to increase the success rate of VS, the different protein conformations should be taken into account in spite of the increasing computational cost. The above-described obvious complementarity of the ligand and structure-based methods led to various combined, integrated approaches over the last 10 years [10,11,12].

## 2. The Major Principles of the Integrated Virtual Screening Approaches

The most common strategy is the *sequential approach* that divides the VS pipeline in consecutive steps (Table 2). It normally starts with a prefiltering 2D similarity search to reduce the virtual chemical space to a manageable number (several thousand) for the more computationally demanding SB methods, that are generally used at final stages on a small number of compounds. This approach was successfully used for many different targets.

In the *reverse sequential approach,* SB VS is applied first, followed by 2D similarity searching using the hits as reference molecules selecting compounds, that are similar to the newly identified hits (hit expansion) [13]. The reverse sequential integration of structure and similarity-based VS method has the following sequence [14]: (A) ligand docking is used to identify an active compound; (B) a pharmacophore or fingerprint method is used to generate a mathematical representation; this is used to (C) enrich a ligand library with compounds similar to the initial active compound.

In the *parallel approach*, both LB and SB methods are run independently and the best candidates identified from each separate method are selected for biological testing. It requires a careful selection or the development of a consensus score (data fusion) that provides a single ranking of the virtual hits.

The *hybrid method* comprises approaches that represent a true combination of LB and SB techniques into a standalone method. There are two main approaches: (a) interaction-based methods (pharmacophoric features and quantitative structure-activity relationship (QSAR) models) and (b) similarity-docking methods (combination of molecular similarity and docking techniques) [12].

Selection of the optimal combination strategy depends on the target, target information, available active compounds, etc.

Integrated VS approaches often use various machine-learning (ML) algorithms. Machine learning (ML) is extensively used to teach the computers how to handle the data more efficiently and ML-supported VS enables medicinal chemists to efficiently find potential lead molecules among millions of compounds increasing the success rate and accelerating the discovery process [15]. Several recent reviews describe the most popular ML algorithms and their contribution to the performance of VS, including Naïve Bayesian classifiers, k-Nearest Neighbors, Support Vector Machines, Random Forests, and Artificial Neural Networks [16,17].

LBVS with 2D fingerprints are particularly well suited to be used as training data for machine learning algorithms [18].

Various recent integrated examples are shown in Table 3.

## 3. Integration Involving In Silico and In Vitro Screening. Our Approach

A workflow has been developed with a first round in vitro biological screening and a second round hit refinement.

In vitro biological screening involved 2D ligand similarity based focused libraries. The second round hit refinement (validation) library was generated by a sophisticated (sequential or parallel) integration of various additional techniques such as 3D ligand- and structure-based modelling, pharmacophore matching, based on the activity results of the first round. The rationale behind this workflow is that (a) in many cases, VS methods and in vitro HTS were combined [30] and found complementary to each other [31,32]; (b) 2D similarity search allows us to select potential candidates from multimillion commercial libraries, providing compounds with reasonable diversity for testing (c) after the first round screening based on the acceptable number of hit structures the 2D similarity approaches could be combined (sequentially or parallel) with more computation demanding 3D ligand or SB methods in order to increase potency and/or structural novelty.

Figure 2 displays the typical workflow we applied in many instances.

Note, a reverse (non-conventional) sequential approach is also involved in vitro screening as part of the screening cascade. Wang et al. [33] intended to identify potent glycogen synthase kinase 3alpha (GSK-3alpha) inhibitors. Based on the related isoenzymes first a homology model was generated then 300,000 compounds were docked. The best scoring compounds (47) were subjected to radiometric assay and 9 hits were identified. Finally, the 2D similarity search was conducted using the two best compounds to fetch more commercially available compounds containing the same scaffold. The major difference between our and the so-called reverse sequential strategy (e.g., applied by Wang, above) is that after in vitro screening we intended to improve the novelty (expanding to additional chemotypes) and validate the 1st round screening, whereas the other methods attempt to expand the chemical space around the hits, most likely retaining the identified chemotypes. Both approaches are reasonable, and they depend on the quality of the target information, available known actives, as well as the available software and computation. With the rapid development of these techniques and the quality of the available target structures the ratio of the reverse integrated approaches will further increase; however, the simple and rapid, initial 2D similarity selection will remain popular, readily available and easy to use for the conventional medicinal chemists.

### 3.1. Elements of the 2D Ligand Similarity-Driven Virtual Screening

*The similarity principle.* The key concept of the LB VS approaches [34,35] is the Similarity Property Principle, which states that similar molecules should have similar biological properties [36,37]. In other words, a molecule that has not been tested for biological activity but that is structurally similar to an active molecule (reference or seed molecule) is also likely to be active [38].

There are various similarity perceptions and concepts [36]. For example, *1D chemical similarity* denotes the similarity of the physico-chemical properties (this property or parameter space of the ligand active on the particular protein target could serve as a pre-filtering tool of the virtual libraries); *2D molecular similarity* focuses primarily on the structural features (e.g., shared substructures, ring systems, topologies, atom connection/path etc.), while *3D similarity* includes shape, flexibility, conformation and stereochemistry. Furthermore, similarity could be global or local. For example, pharmacophore models represent a local view of similarity focusing only on selected atoms, groups, or functionalities that are expected to be responsible for activity. Finally, *biological similarity* means similar activities regardless of the chemical structures.

The 2D similarity principle is generally valid for many targets, when the new molecules bind in a similar fashion to the target receptor. However, “chemical similarity is not a foolproof guaranty for a common action mechanism of all congeners” (Scior, T. [39]). Thus, for several targets the ligand and biological similarity (searchable ligand space and the activity islands) is just partly overlapping. The size and continuity of the activity islands are very important. Activity landscape and cliffs could certainly limit the success. Activity cliffs are present when small changes in compound structure (having high similarity coefficient values) lead to great changes in activity [40]. It highlights the importance of screening high density combinatorial libraries matching the “cliff-rich” regions of activity landscapes [41].

Bajorath et al. [42] investigated the similarity versus potency distribution for several targets and found varying distribution of potent inhibitor pairs over the similarity expressing Tanimoto coefficient (T2D) range. For FactorXa, T2D was between 0.2 and 1, which reveals a broad diversity of the active compounds; therefore, the similarity search could become challenging.

There are two major elements required for 2D similarity search. *The searchable chemical space* comprises the commercially available unique structures. The number of compounds are continuously increasing over the years (Table 4 upper portion). *The biologically relevant chemical space* represents the “source (reference or seed) compounds”. Several annotated databases provide biological activity data and target information for large number of small molecules (Table 4 lower portion). Bibliographic search on the particular targets and ligands could complement the databases.

*Software and 2D fingerprints*. The most common molecular (2D) similarity search uses 2D fingerprints, i.e., binary strings encoding the presence or absence of a substructure within the molecules [50]. Applying simple 2D fingerprints is often the method of choice [51], particularly when numerous reference compounds and multimillion compound databases are available. The theory of the major fingerprint types (substructure-based, topological or path-based fingerprints such as chemical hashed fingerprints, circular fingerprints such as Extended Connectivity Fingerprint—ECFP, and pharmacophore fingerprints) are extensively discussed in the literature [6,9,40,52,53].

*Similarity measures*. [37] In order to quantify similarity, pairs of molecular fingerprints are compared and the ratio of the common and unique substructures (fingerprints) is determined en masse. Most frequently, Tanimoto coefficient [54] is used for measuring similarity [55] in spite of its marked size-dependency. (Note: in this case Tanimoto coefficient refers to 2D Tanimoto similarity with abbreviation of T2D. If Tanimoto similarity is applied to measure 3D similarities it is labelled as T3D). T2D typically yields low similarity values when the reference molecules are relatively small and yield narrower T2D value distributions for more complex compounds [53].

The origin of the ‘mythical’ T2D = 0.85 comes from an early study by Martin et al. using chemical hashed fingerprints. Analyzing more than 100 biological screening data, the T2D threshold value of 0.85 was found to correspond to approximately 30% probability that a database compound which resembles an active compound was found to be similarly active [38,53]. While many experts state that there is no universal cutoff value and it varies target to target [40], the most widely used T2D threshold value is ~0.65, even though it corresponds only to a statistical similarity at *p* = 0.01 [36]. On the other hand, such relatively high structural similarity does not necessarily lead to novel chemotypes or scaffolds, which is one of the major challenges in the contemporary drug discovery. It is widely discussed that new chemotype discovery or scaffold hopping is possible if we go below T2D = 0.65 similarity value [56,57]. Lower similarity thresholds also afford higher structural diversity, which might lead to novel chemotypes with or without scaffold hopping even though the hit rate is somewhat reduced [58]. It is important to note that while the Tanimoto coefficients of using chemical hashed and related connectivity-based fingerprints are comparable (as used in all cases of the present account), substructure or pharmacophore-based fingerprints require different thresholds for similarity selection. For 3D similarity searches the determination of the appropriate Tanimoto cut-off values requires individual studies since it is strongly dependent on the applied algorithm as shown later under Section 4.1.

Group fusion methods apply multiple reference compounds and combine the individual similarity measures [59]. In this approach the similarity scores are first calculated to ‘n’ reference compounds for each compound within a medium sized searchable compound library and then the ‘n’ similarity measures obtained for each compound of the library are accumulated (‘fused’). The calculation is followed by ranking the compounds according to the ‘fused’ similarity measures.

Determining the similarity between a biologically active reference compound and each molecule in a searchable chemical database is followed by ranking the database molecules according to the similarity scores. Such compounds that are above the pre-set similarity thresholds constitute an initial 2D focused library subjected to multi-step in silico filtering.

### 3.2. Filtering the Initial 2D Similarity Search Results

*Physico-chemical parameter and property space filtering.* The empirical physico-chemical parameter ranges stand for drug-likeness collected in Lipinski’s Rule of 5 [60] and Veber rules. If most of the parameters of the drug candidate fall into the pre-defined ranges the concerning molecule could be administered orally [61,62]. Virtual focused library generation is often linked to target families that represent a distinct chemical, biological, and property space. Each target or target family has unique parameter/property ranges (space) that are different from the above rules [63]. Thus, calculation of the physico-chemical parameters for the known active compounds, the preferred ranges could be determined. (Practically, the middle, 90% of the ranges were applied omitting the outliers). Note, the similar physico-chemical parameter ranges represent the first level, 1D (chemical) similarity as shown before. The above rules and/or the predicted target-specific property cut-off values allowed filtering the initial selected library.

*Diversity and related selections.* If the focused library is still huge (too many high similarity (high T2D) compounds obtained), diversity selection, clustering, or scaffold analysis are the methods of choice to reduce the number of compounds for biological screening. Diversity selection [64] could follow the in silico property-based filtering, which results in a more diverse molecule library, where the structural clusters or molecular scaffolds are evenly represented. Clustering is usually based on the Maximum Common Substructure search [65] algorithm. Scaffold analysis could also be combined with the simple chemical diversity selection and a combined index could reflect both (EDI, Explicit Diversity Index) [66].

## 4. Case Studies for Integration of 2D/3D In Silico/In Vitro Approaches

### 4.1. Sequential Combination: 2D/3D Ligand-Based Similarity Approaches and In Vitro Screening: PDE4/PDE5 Inhibitors 

For better and more efficient performance the 2D/3D similarity selection methods can be combined. Thus, we analyzed the rational for creating a fusion (T2D/T3D) score and a reasonable cut-off value [67]. The possible binding features of small molecules can be assessed by their conformational flexibility and shape. Applying flexible alignment (such as Screen3D software, ChemAxon, Kalászi et al. [68]), the statistically average conformations (generated by the accumulated dynamic 3D structures) can be compared, and, based on that, the similarity between two compounds could be assessed and characterized by 3D similarity measures (3D Tanimoto = T3D).

#### 4.1.1. 2D/3D Similarity Correlation Analysis of Previously Generated PDE5 Focused Libraries

In the correlation study we involved 5 PDE5 inhibitor hit compounds, identified in a previous study [69], selected based on their 2D similarity to three seed compounds using 5 M commercial vendor libraries (IC_50_ < 10 µM, Table 3; the cut-off value was T2D ≥ 0.6 for the 2D similarity selection). Their 3D similarity values (using Screen3D) were generated towards the same seed compounds.

The T3D values were more sensitive to the structural differences and were much lower than the T2D values (Table 5). In order to extend the study, we calculated the 3D similarity values for all the compounds we obtained from the 2D similarity selection to the three seed compounds with T2D ≥ 0.6 selectivities and the 2D/3D correlation is shown in Table 5. We found that while the 2D selection criteria was T2D ≥ 0.60, the 3D Tanimoto scores (T3D) were above 0.2, since the T3D values for the five hits were above 0.3, even though many “2D similar” compounds fell into the 0.2–0.3 category.

We could conclude that, even though the compounds selected by 2D methods had similar molecular architectures (atomic connectivity), they are rather different in terms of shape and conformational flexibility; therefore, the lower T3D region suggests that such 3D “dissimilarity” represents significantly different binding characteristics.

It is also interesting to correlate the 3D similarity values with the biological activity of the same cluster (41 compounds) (Figure 3). While the five identified hits (exhibiting ≥ 57% PDE5 inhibitory activity at 10 μM concentration) have T3D values higher than 0.3, there are lots of 2D similar compounds that has 3D similarity values above 0.3 but they are not hits.

After the first round selection and screening we had the conclusion that T3D = 0.3 might have been a useful cut-off value even if it might result in numerous virtual false positives.

The 3D similarities were calculated for the second round screening dataset (104 compounds, generated by 2D similarity search around the five first round screening hits using a higher selection criteria, i.e., T2D ≥ 0.8, with the same commercial library). While the T2D cut-off value was 0.8, the corresponding T3D values were between 0.35 and 0.95 (Figure 4). The majority of the compounds have T3D values above 0.6.

Correlation of the IC_50_/T3D values for 17 s round hits derived from five first round hits (in separate color) is shown on Figure 5. (Note: 2D similarity is over 0.8 for all the compounds according to the selection criteria and the hit criteria: IC_50_ ≤ 2 μM).

As a result of the above analysis we made the conclusion that T3D values are much more sensitive measures than their T2D counterparts, and they reflect important structural features (e.g., conformational flexibility). We could also make the statement that T3D = 0.3 would be a good cut-off value in the first round similarity selection and T3D = 0.5 or 0.6 in the second round (hit validation). We could get probably higher hit rate applying stricter filters, but then some compounds (chemotypes) would have been lost.

The T2D and T3D similarities of the second rounds screening library (104) towards the corresponding first round hits were between T2D = 0.8–0.98/T3D = 0.35–0.95, and for the hits (17) were between T2D = 0.80–0.93; T3D = 0.47–0.91.

We attempted to combine the T2D/T3D similarities into a single fusion score: T2D + T3D. The T2D/T3D values would range between 1.28 (0.8 + 0.47) and 1.83 (0.92 + 0.91) based on the similarity scores calculated for the second round hits towards the first round hits.

Summarizing the above findings, we concluded that combination of the 2D similarity search with 3D similarity measures might be a useful and applicable approach in ligand-based virtual screening and helps to successfully refine 2D selections.

#### 4.1.2. Sequential (or Fused) Application of 2D/3D Similarity Methods, Selection of PDE4B Libraries

For first round selection and screening, the typical 2D similarity search was carried out using 44 known PDE4 inhibitors as seed compounds and the 5 M compound vendor repository as the drug-like chemical space. T2D ≥ 0.65 and property space filtering and diversity selection and visual review resulted in 105 available compounds. In vitro screenning (at 10 μM concentration) identified nine hits (inhibition > 47%). Seven compounds had IC_50_ values between 0.05 and 16 μM. It recovered the same chemotype in five cases, with minor scaffold hopping.

For the second round (hit validation) library selection, the input structures were the seven first round hits. First, a standard 2D similarity search was executed on the same commercial library, because Screen3D could only handle a couple of thousand compounds rather than a couple of million. After property space filtering 1341 compounds were obtained (T2D > 0.65 was applied instead of T2D > 0.8 for covering a bigger chemical space).

In the next step we calculated the 3D similarity values for 1341 compounds towards their corresponding first round hits. Their correlation to the 2D scores showed a similar trend as with PDE5 (the 3D Tanimoto coefficient, T3D ranged from 0.19 to 0.99 while the 2D values were ≥0.65), with a weak, linear correlation between the 2D and 3D similarity measures.

The fusion score was applied to refine the 2D similarity search in a second case study where we aimed at selecting and evaluating a PDE4B focused library.

We decided to use a fusion score which handles the influence of T2D and T3D equally and aggregates the scores. A suitable cut-off value was applied as T2D + T3D ≥ 1.5 based on the previous analysis of the PDE5 data. The workflow of applying the fusion score in the second round focused library generation is displayed in Figure 6.

Applying this combined cut-off value (T2D + T3D ≥ 1.5), the library size was reduced from 1341 to 233. The library was further reduced to 70 by diversity selection and 35 compounds were received from vendors. The compound set reflected the distribution of the initial set of compounds (233). In general, the contribution of T2D is gradually increasing but fluctuating between 0.65 and 1, while T3D is decreasing from 1 to 0.6 among the compounds that have a fusion score of 1.5.

The application of this fused T2D/T3D similarity measure led to an increase of the hit rate from 6.8% (first round, 47% inhibition at 10 μM, <2 µM = 2.8% (three compounds) to 28.5% (second round at 50% inhibition at 10 μM) and the best two hits had 53 nM inhibitory activities.

In the second round no additional chemotypes were found, but it strongly expanded the chemical space around the first round hits (Figure 7).

**Conclusion:** the T2D/T3D fusion score contributed to increase the hit rate significantly within the dominating chemical space. (Note: T3D is much more sensitive to the structural changes and well correlates better to the biological activity). The hit rate of the second round screening was 28.5% (>50% inhibition, at 10 μM), while 10% hit rate was calculated if only those compounds were considered that had an IC_50_ values < 2 μM (seven compounds).

### 4.2. Bioisoster Extended Sequential Combination of 2D Similarity/Pharmacophore Model Search and In Vitro Screening: Inhibitors of a Serine Protease, the Complement Component C1s 

In the first round routine, 2D similarity search was carried out in order to identify C1s inhibitors with novel chemotypes. After clustering the various previously reported inhibitors that have low μM inhibitory activities, and removing the redundant structures, eight amidine/guanidine and 12 non-amidine/guanidine compounds were selected as seed/reference compounds for 2D similarity search [70]. Since most of the known C1s inhibitors contain amidine/guanidine moieties, bioisosteric replacements for those substructures were also involved in the reference structures. For generating the bioisosteric alternatives we applied literature sources and databases. The two searches resulted in 4667 compounds applying T ≥ 0.65 Tanimoto coefficient as a similarity cut-off value (Figure 8). Thus, the initial C1s focused library was filtered for the above physico-chemical parameter space of the known inhibitors leading to 206 structures for first round screening. After diversity selection and visual inspection, we purchased 50 compounds, which contained 1,2,3-benzotriazoles, 1,2,4-triazoles, imidazoles, 3,1-benzoxazin-4-ones, that reflected known non-amidine chemotypes plus a number of unrelated (novel) cores coming from the bioisosteric replacements. Biological screening led to a relatively high hit rate: 22% (the inhibitory activity was <10 μM); however, it did not reveal novel chemotypes. The hit structures were closely related to the seed compounds but revealed new analogues with different substituent pattern and improved C1s inhibitory activities. The two highest activity compounds belonged to the 1,2,4-triazole cluster (IC_50_ = 12 and 44 nM).

In order to identify C1s inhibitors with novel chemotypes we turned to building pharmacophore models, while 3D SB methods were found inappropriate (Figure 8). Three major chemotypes were involved in the model building that included both the literature (16) and in-house identified (10) active compounds.

*Searchable chemical space.* Physico-chemical parameter space filtering of the 5 million compound vendor libraries and removal of the duplicated structures resulted in 445,457 compounds. In the next stage, phase database creator was used to develop pharmacophore libraries leading to 679,420 structures that were used as entries in the pharmacophore-based VS. Compounds with the highest 50 phase scores were selected for the three compound clusters. The VS was carried out matching the best pharmacophore models selected for each cluster models 1,2,3-benzotriazole (Model: AARR_3), 1,2,4-triazole (Model: AHRR_1), 3,1-benzoxazin-4-one (Model: AARR_3), representing the 2D focused library screening hits plus literature compounds (Figure 9).

The pharmacophore based screening resulted in six active compounds out of the purchased 21 (hit rate = 28.5%). The library contained 1,2,3-benzotriazoles; imidazoles, 3,1-benzoxazin-4-ones as known chemotypes, while the rest belonged to previously not identified chemotypes among the C1s inhibitors. Two compounds that showed inhibitory activities belonged to the novel chemotypes: one contains a thieno[2,3-d][1,3]oxazin-4-one core (IC_50_ = 549 nM) and the other (1,3-benzoxazin-4-one, IC_50_ = 241 nM) can be considered as a bioisoster of the known “reverse” 3,1-benzoxazin-4-one chemotype. The remaining hits were 3,1-benzoxazin-4-ones and benzotriazoles.

**Conclusion.** 2D similarity search did not lead to novel chemotypes, however, pharmacophore model generation allowed us to identify two novel chemotypes with sub micromolar activities.

### 4.3. Parallel and Sequential Combination of 2D Similarity/3D Docking and In Vitro Screening: PDE5 Inhibitors

The first round library generation relied on 2D similarity search applying a diverse set of known PDE5 inhibitors (27), and a 5 M commercial library was used as the targeted chemical space (Figure 10) [69]. The LB 2D similarity search was performed by setting a similarity threshold (>0.60 Tanimoto) and the most similar compounds (50–500) were iteratively selected for each seed compound starting with 0.75 Tanimoto coefficient (T2D). If the number of virtual hits were between 50 and 500, it was accepted, while if the number of virtual hits was <50 or >500, the similarity level was decreased or increased by 5% (or 2.5%) until the optimal range (50–500) could be achieved. In some cases where only few compounds were identified the similarity threshold was allowed to go down to 0.60 Tanimoto coefficient. The hits were related to the reference compounds.

The first round in vitro biological screening resulted in eight compounds showing inhibition >55% at 10 μM concentration (hit rate: 8.2%), and two compounds showing IC_50_ values below 1 μM. All eight hits represented different chemotypes, and interestingly, derived from six seed compounds.

In all cases the reference chemotypes (scaffolds) had slightly or significantly been changed leading to “scaffold hopping”. The structural motifs of the reference compounds were reflected in the eight compounds; however, in many cases in a virtually “rearranged” manner (see Figure 11).

For the second round of VS, six compounds were selected as seeds based on novelty and efficacy (Figure 10). First, 2D similarity search was performed using the searchable chemical space (as applied for the first round, ~5 M compounds) at >0.8 Tanimoto coefficient (resulting in 849 compounds) and for second round biological screening we have chosen the most similar 10 compounds and another 10 compounds as a result of a diverse selection of the similar compounds at the above similarity level. For four first round hits (chemotypes, #2, #3, #6, #4) we obtained analogues with higher efficacy.

For sequential 2D similarity/3D docking approach, first 1810 analogues were selected at >0.75 T2D by 2D similarity around the six first round hit compounds, applying the searchable chemical space (~5 million compounds). The selected compounds were docked to the PDE5 active site (PDB = 1 TBF). The compounds were then arranged according to the fitting scores and the first 60 compounds were selected, 52 were different from the above second round 2D similarity search set, and 48 compounds were available for purchasing. Interestingly, the analogues that were most similar to the first round hits and were active in biological assays did not necessarily showed high fitting scores.

The second round selection and screening resulted in higher hit rate than in the first round. Even though the pure 2D similarity showed slightly higher hit rate than the combined 2D/3D approach the latter method increased the diversity of the hits and led to more active hits (Table 6).

The three active compounds coming from the 3D docking of the 2D similarity search (having efficacy below 1 μM) representing the novel first round chemotypes showed higher efficacy compared with the pure 2D second round selection. In many cases the similarity to the active reference (seed) compounds dropped below T = 0.60 (as applied in the first round) and went even lower; thus, a lower similarity higher activity was achieved (Figure 11).

Further hits (11 out of 48) out of the 3D docking set were not identified as close analogues of the first round hits by 2D similarity approaches; however, they had high docking scores instead.

**Conclusion:** these results propose that 2D and 3D methodologies are complementary and their combination would readily improve the performance of the in silico screening.

### 4.4. Parallel Combination of 2D Similarity/Pharmacophore Model Search and In Vitro Screening: 5HT_6_ Antagonists

First, the reference space was defined by collecting 49 known 5-HT_6_ antagonists (‘seeds’) representing various chemotypes (cca. 25) from available literature [71].

After applying the routine 2D similarity search protocol (T2D ≥ 0.65), property space filtering, diversity selection, and visual inspection, 91 compounds were acquired and screened in the first round. Compounds (12) having antagonist activity >85% (at 10 μM concentration) were considered as hits and selected for IC_50_ determination. Six compounds (out of 12) had IC_50_ values ≤ 1 μM. As Tanimoto (T2D) values are between 0.66 and 0.77, significant diversity can be seen although the major chemotypes have not varied much; thus, low structural novelty was achieved, while the hit rate was 13% and from the architecture of the parent (seed) compounds we could recognize the selected compounds.

The second round library generation we applied two separate approaches (2D similarity search and pharmacophore model matching) in a parallel manner (Figure 12). For the pharmacophore model generation we used 49 structurally different seed compounds and 11 first-round hit molecules. In order to increase the coverage by pharmacophore models, we divided the molecules into three distinct clusters, which allowed to create a pharmacophore hypothesis with five sites for these three datasets (cluster 1: AAHRR type; cluster 2: AADRR type; and cluster 3: AAHRR type). The five-point pharmacophores contain two hydrogen bond acceptors (A), one hydrophobic group (H) or hydrogen bond donor group (D), and two aromatic rings (R). Cluster 1 contained 12 seed molecules and 5 hits; Cluster 2 had 11 seeds and 6 hits, while cluster 3 (sulfone-bridge) contained only 12 seed molecules and no molecules from the first round hits.

All three pharmacophore models were used to screen a combined vendor database which contained over 2.2 million vendor compounds. We checked manually the 50 top scoring molecules coming from each pharmacophore model searching (altogether 150) and we bought 70 of them from the three compound suppliers. Sixty of them were received and tested for 5-HT_6_ receptor antagonism.

In comparison, we also carried out a typical 2D similarity search using the six first round hits (IC_50_ ≤ 1 μM) as seed compounds and the same 2.2 M vendor library was used as drug-like chemical space similarly to the pharmacophore-based in silico screening. As a cut-off value T = 0.8 was used, and 60 compounds were gained after property- and diversity-based selection and ordered. Finally, 56 compounds were received.

In our experience, pharmacophore search was found to be a particularly powerful tool for selecting compounds from large size compound libraries: 20 compounds of 50 were found active (>85% antagonist activity, hit rate = 40%). The average 2D similarity to the original first-round seed compounds was 0.586, which has already presumed novelty, and indeed, three novel chemotypes were found compared with the seeds. The average biological activity of the hits was IC_50_ = 467 nM (most active hit—IC_50_ = 15.3 nM). Pharmacophore matching also lead to three novel chemotypes compared with the seed compounds used in the first round similarity search (Figure 13). Two novel heterocylic ring systems were particularly interesting and showed high antagonist activity. The closest analogue in the seed set is shown in Figure 13, which also revealed that T2D similarity values were less than 0.65; therefore, it could not be retrieved during the first round similarity search.

Two-dimensional similarity search (T2D ≥ 0.8) around the structure of the six first-round hits (IC_50_ ≤ 1 μM) gave better results in terms of the hit rate (27 compounds out of 56 were found active, hit rate = 51%, most active hit—IC_50_ = 1.9 nM), and the average 2D similarity values to the original seeds were 0.704 (at that level, little novelty was expected).

In summary, pharmacophore search led to novel chemotypes compared to the seed structures, while 2D similarity search had a slightly better hit rate but lacked novelty.

**Conclusion:** The integration of the 2D similarity search with pharmacophore matching looks like a powerful (complementary) combination for selecting focused libraries with enriched novelty. While 2D similarity search is useful for hit validation, pharmacophore matching is valuable for extending the activity space with novel chemotypes.

### 4.5. Parallel Combination of 2D Similarity Search/Fragment-Based Design, 3D Docking, and In Vitro Screening: Glutaminyl Cyclase (QC) Inhibitors

Two strategies were applied to identify potential fragments from commercial drug-like and fragment libraries (Figure 14).

Strategy I: Selection of the fragment library via direct 2D similarity search [72]. Collection of potential active fragments toward a biological target starts from the structure of known biologically active compounds (active chemical space), carrying out a 2D similarity search using multi-million compounds libraries (T ≥ 0.65) and then filtering the resulting analogues for physicochemical parameters reflecting the fragment criteria. A modified Rule-of-3 was applied (Mwt < 250, cLogP < 3.0) which was slightly stricter than the standards widely applied, in order to provide acceptable number of fragments. The united fragment library (204 compounds) was purchased, pre-screened by biophysical methods (DSF), and the hits were further investigated by functional in vitro assays.

Strategy II: Selection of a fragment library via fragment disconnection, bioisosteric replacement combined with 2D similarity search. Fragment disconnection of 15 known QC inhibitors resulted in 19 fragments that were applied first in 2D similarity search on five vendor fragment libraries (500,000 compounds) at T ≥ 0.85 similarity level.

Ten compounds showed acceptable inhibition, and 9 out of the 10 structures were arranged into three clusters based on similarities. The best hit inhibited QC with IC_50_ = 12 μM. The overall hit rate was ~5% (considering the 10 fragment hits). In the fragment-based drug discovery the identified fragments are either grown to drug-like molecules by additional groups to fill the binding cavity or two fragments could be linked depending on their location within the active site. In order to get better insight about the binding interactions of the fragments, we have docked the identified fragments into the three-dimensional structure of QC (PDB = 3 si0). Docking calculations suggest that two fragments marked appear to bind to complementary parts of the binding site, thus, it allowed us to connect those fragments to occupy jointly the binding site (Figure 15, linkage is marked in red).

Two-dimensional similarity search was used to select similar compounds to the best scoring virtual linked fragment from multimillion vendor databases. The similarity search resulted in 77 compounds with relatively low, but still acceptable similarity score (T = between 0.55 and 0.6). The 77 similarity hits were subjected to docking and we created a list of the best 10 molecules based on the flexible MM-GBSA calculations. Such compounds represent a small drug-like focused library for QC inhibition (Figure 16).

**Conclusion:** A 2D similarity search is suitable to support fragment-based drug discovery. The identified fragments are allowed to generate virtual linked fragments which could serve as a seed compounds for combined 2D similarity/3D docking strategy to identify potential novel QC inhibitors.

### 4.6. Parallel 2D Similarity Selection/Group Fusion Aproach, Combination with In Vitro Screening: Melanin-Concentrating Hormone Receptor-1 (MCHR1)

For the first round screening the typical 2D similarity protocol was used. The reference (seeds) compounds’ set was composed of 43 structurally diverse known, active MCHR1 inhibitors representing 19 chemotypes [73]. The searchable drug-like chemical space was composed of 5 million compounds derived from 10 top vendor libraries. The similarity level was set as Tanimoto ≥ 0.65. Property space filtering, diversity selection, and visual inspection followed the similarity search. In the first screening rounds, 48 compounds showed ≥70% MCHR1 antagonist activity at 10 μM concentration. Among them, 11 compounds showed MCHR1 antagonist activity at IC_50_ = 2.3 μM and below, which corresponds to a 4.2% hit rate, and 5 compounds showed higher activity than 1 μM (1.9%).

For second round (hit validation/extension) library generation the 2D similarity selection was done first applying Tanimoto ≥ 0.70 as a cut-off value. This resulted in 4832 compounds from the same commercial chemical space as applied in the first round, without taking into consideration the property ranges (Figure 17). The 10 most similar compounds identified to each first round hit (second round seed) were selected and this list formed the first portion of the second round focused library (119 compounds).

In parallel, group fusion approach was applied as an extension of the simple 2D similarity selection. In this approach not only the highest similarity scoring compounds were considered, but the Tanimoto values that calculated between each compound and all the reference compounds were added together (generating a fusion score) and finally the compounds were ranked according to such values in a decreasing order. The group fusion calculation and ranking was carried out on a smaller set obtained after the same initial set was reduced to 2239 compounds by property space filtering. Ten highest group fusion scoring compounds per seeds formed the group fusion-based focused library (169 compounds without overlaps) and then merged with the highest 2D similarity compounds forming the MCHR1 antagonist focused library (242 compounds after removing the overlaps) that was acquired. Finally, in the second round screening 21 compounds were able to antagonize the receptor at (IC_50_ ≤ 1 μM, hit rate: 8.6%), and 17 compounds were active below 500 nM (hit rate: 6.9%), which constituted a significant improvement. Interestingly, the best three hit compounds resulted from the group fusion selection (Figure 18).

In principle, compounds with high group fusion score merge many structural motifs of the seeds into one single compound since they show similarity to structurally diverse seeds. Such approach is particularly useful for the MCHR1 antagonists which are typically linear compounds connecting various structural motifs. The group fusion approach helps to select compounds that contain such motifs that are responsible for the activities, often in novel connections or arrangements. Such compounds may not be retrieved by the simple 2D similarity selection (Figure 18).

**Conclusion:** Group fusion provides an extension and refinement compared with the single seed compound drive 2D similarity search. This method is particularly powerful if the typical bioactive compounds have mostly a linear arrangement. In that case the group fusion method helps to identify the preferred bioactive fragments (motifs) in a favored arrangement.

## 5. Summary, Conclusions, and Additional Application Areas

Rapid 2D similarity search can be performed on multimillion compound’s databases if structures of active molecules are available; therefore, it is a “real time” method compared with the other LB approaches (including shape-based screening—3D flexible alignment or pharmacophore building). It requires minimal computational setup and configuration as well, and conventional medicinal chemists could easily use and interpret the data. Furthermore, the broad availability of commercial compound databases and large annotated libraries containing molecular structures and biological activity data make this approach often as a first choice, and a filtering tool reducing the number of candidates for more computational intensive 3D ligand and SB VS methods. On the other, behind the speed and easy handling there is a marked simplification: 2D does not require pharmacophore hypotheses or specific knowledge about activity-relevant features of the compounds and ignores the shape and stereochemistry. Furthermore, 2D is a flat representation of the structures with binary fingerprints, which helps the rapid comparison of the reference and target molecules and ranking based topological, atom connectivity-based similarity. Despite such marked simplicity, the 2D similarity search methods often provides better VS performance than 3D shape-based approaches [74,75].

The performance of the 2D similarity search relies on the diversity of the seeds and the searchable chemical space (vendor libraries), the computational methods (fingerprints) as well as the applied criteria (similarity cut-off values, property space filters, etc.).

In cases where the 3D structure of the target is unknown or its prediction by SB methods is challenging, 2D methods are preferred. Such an example is the discovery of novel lysyl oxidase isoenzyme (LOX, LOXL2, LOXL3, LOXL4) inhibitors, where apart from the above challenges only a limited number of known active compounds were available as well. In that case the reference list was extended with bioisosteric analog structures. Two-rounds 2D similarity search led to the identification of six compounds that inhibited any of the LOX family enzymes <10 μM concentration [76].

The above example is not typical, as in most of the cases using molecular fingerprints and 2D similarity search is one part of a more complex VS workflow rather than a single standalone method. In the described case studies, 2D similarity search was applied not only a filtering method but the search results were evaluated by in vitro screening and the first round hits served as starting points for various, more focused 2D/3D integrated approaches, leading to improved hit rate and hits with higher activities and often resulted in novel chemotypes.

The combination with other mostly 3D ligand and SB methods is well-justified since 3D information (stereochemistry, shape, and conformation required for binding etc.) is missing from the 2D similarity search methods. The 2D-based algorithms are usually less accurate but faster than 3D based.

On the other hand, there are certain limitations of the 2D similarity search. There are activity classes that are unsuitable for 2D fingerprint evaluation. The presence of activity cliffs could also have negative impact on the performance.

Furthermore, structural similarity to the reference compounds certainly limits the scaffold hopping capabilities for 2D similarity search methods, although playing with the applied similarity cut-off values (Tanimoto), one could set its preference as increasing the hit rate (high T2D) or the number of the novel chemotypes (low T2D). Similarity VS novelty [77] is thoroughly discussed in the literature. Structural analysis of the virtual and first round screening hits is a necessary task at various stage of the VS cascade. It could include generating a structural evolution tree from the seed to the in vitro hit compounds as well and a novelty check compared with the seeds.

Such illustration (selection of PDE5 inhibitors) contains all the hit structures derived from a reference compound (#18) either in the first/second round screening, together with the similarities to the reference compound. Two hits were derived from the parallel 3D docking. In many cases 2D similarity search is favorably combined with 2D/3D fragment-based approaches, 3D similarity and pharmacophore search methods, as well as 3D docking in order to increase the novelty level (Figure 19). In the above case studies for example pharmacophore model generation and matching the model led to identification of novel chemotypes (C1s, 5HT_6_).

### Additional, Emerging Application Areas for 2D Similarity Search

*In silico identification of protein targets (reverse screening) or off-target effects.* Two-dimensional similarity search offers a cheap tool for the exploration of ligand-target interactions. Chemical similarity search works inversely as well, where based on the structure of a query compound similar compounds are searched from databases having biological annotations (effect and protein targets acting on) to each compound. Based on the similarity principle if the query compound resembles a biologically active compound acting on a certain protein it can be concluded that they share the same target. A typical case if the protein targets should be deconvoluted for the natural products [78].

Similarity-based target identification could be carried out when hits could be paired with specific protein targets after phenotypic, cell-based assays. After a HT differential cytotoxicity screen 2D VS led to the identification of 115 proteins as being hit uniquely by compounds showing selective antiproliferative effects for tumor cell lines [79].

Similarly, off-target effects such as toxicity could be predicted by similarity search between the query molecule and a toxicity database [80,81]. Recently, similarity-based methods were also used in synthesis design assuming that similar compounds have similar reactivities. It was demonstrated that molecular similarity is surprisingly effective for proposing and ranking one-step retrosynthetic disconnections based on analogy to precedent reactions [82].

*Drug repurposing.* Drug repositioning is a process of identifying new therapeutic use(s) for old (existing), previously registered, available drugs. In this process known drugs are searched for databases of compounds that are active on the particular therapeutic areas. A hybrid integrated VS method was described for drug repurposing by including LB, data fusion, and decision tree pipelines [83,84].

After the recent outbreak of the COVID-19 pandemic, intensive drug repurposing efforts have been initiated. In one example a sequential docking/2D similarity hit expansion approach was used to identify existing drugs that likely act on COVID-19 main protease (SARS-CoV-2 MPro) [85].

In another attempt a consensus approach included combining similarity searching with various queries and fingerprints, molecular docking with two docking protocols, and ADMETox profiling. Based on the integrated method they proposed several commercially available compounds (including existing drugs) for experimental testing against SARS-CoV-2 main protease [86].

**In summary**, even though 2D similarity search is an almost 30 year old technique, it is still a method of choice in many virtual screening efforts due to its simplicity, easy interpretation, and relatively good recovery rate. In most of the cases it is not applied alone but in a combination with 3D ligand and structure-based methods. Parallel integrated approaches allowed the comparison of the various 2D and 3D methods and revealed that 2D similarity-based and 3D ligand and structure-based techniques are often complementary; thus, their combinations represent significant improvement in finding active and novel compounds in the early phase drug discovery. In the paper we discussed six case studies including sequential and parallel integration of the 2D similarity approaches with in vitro screening and 3D ligand and structure-based methods. While sequential combination of 2D/3D similarity search increases the hit rate significantly, combination of 2D similarity with pharmacophore model or 3D docking sequentially enriched the resulting focused library with novel chemotypes. In summary, recent integrated approaches could add novel directions to this evergreen virtual screening technique, increasing the success rate of VS.

## Figures and Tables

**Figure 1 molecules-26-05593-f001:**
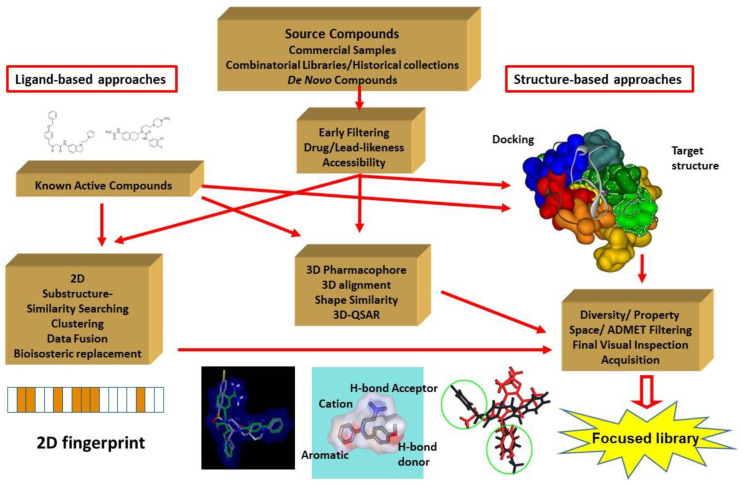
Overview of the virtual screening approaches leading to a focused library for biological screening.

**Figure 2 molecules-26-05593-f002:**
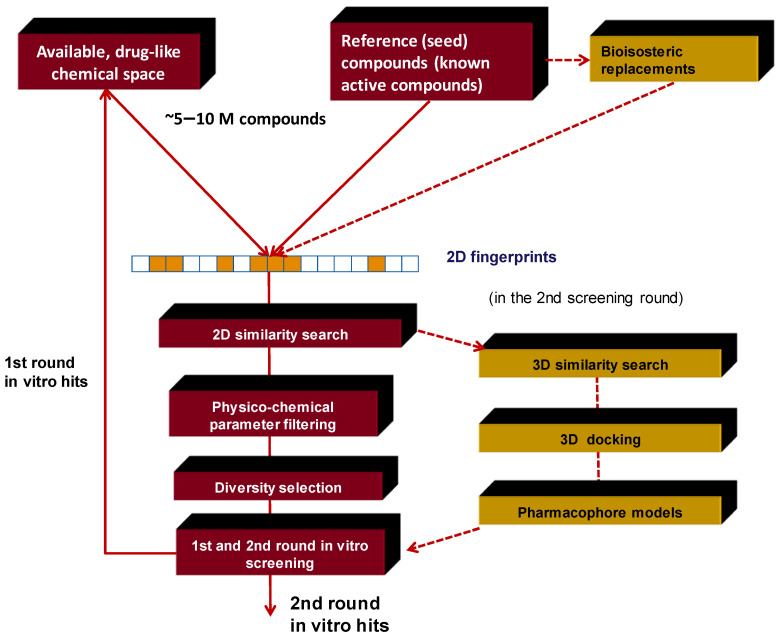
Typical work-flow for multistep virtual screening approach combined with in vitro screening.

**Figure 3 molecules-26-05593-f003:**
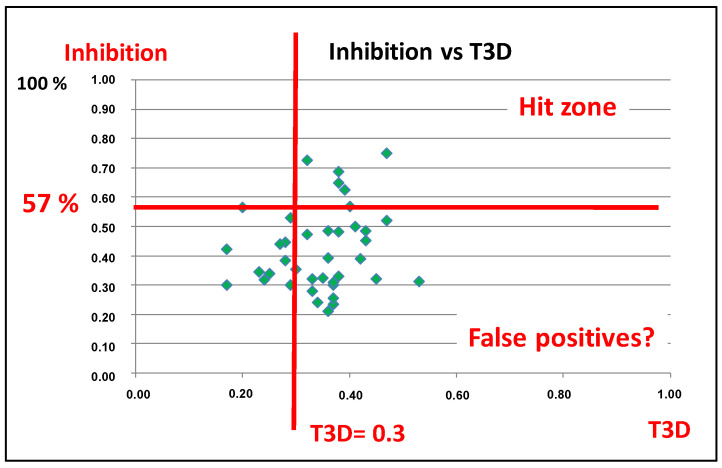
PDE5 inhibition % values were correlated with the T3D values.

**Figure 4 molecules-26-05593-f004:**
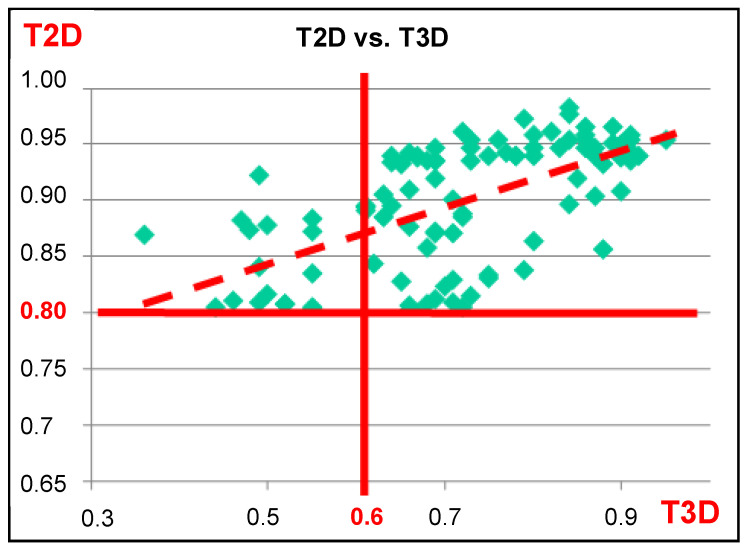
Correlation between T2D and T3D values of the compounds (104) selected for the second round screening.

**Figure 5 molecules-26-05593-f005:**
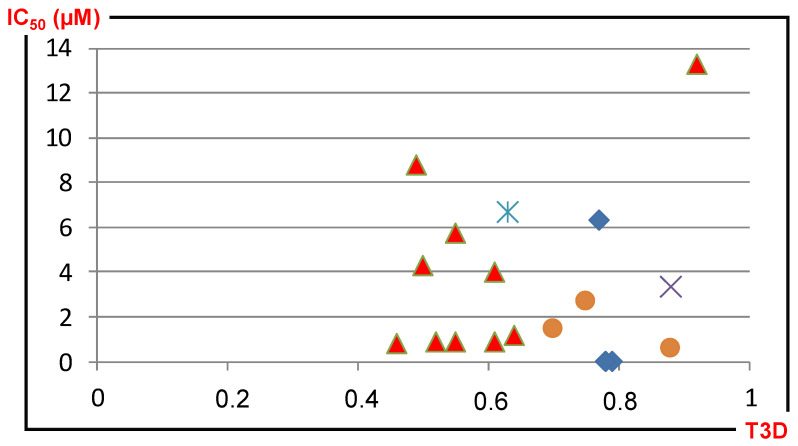
Correlation between PDE5 inhibitory activities and T3D values. (T3D: 3D similarity towards the 5 first round hits, indicated by the different symbols).

**Figure 6 molecules-26-05593-f006:**
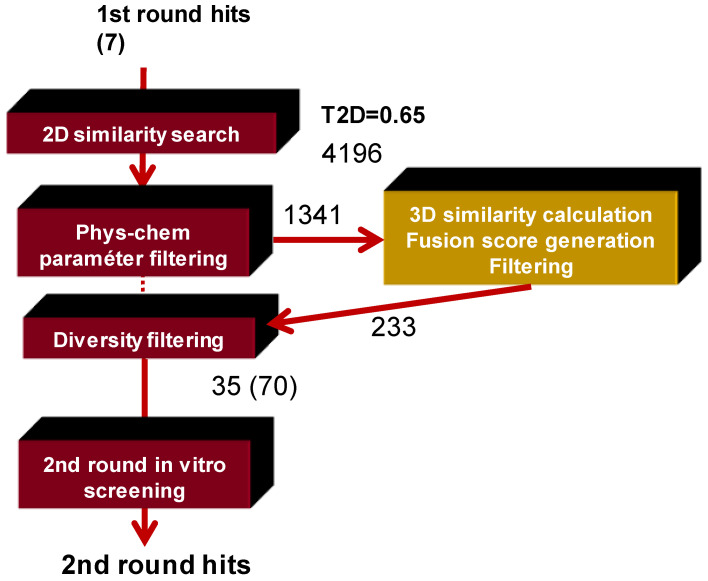
The workflow of applying the fusion core in the second round focused PDE4B inhibitor library generation.

**Figure 7 molecules-26-05593-f007:**
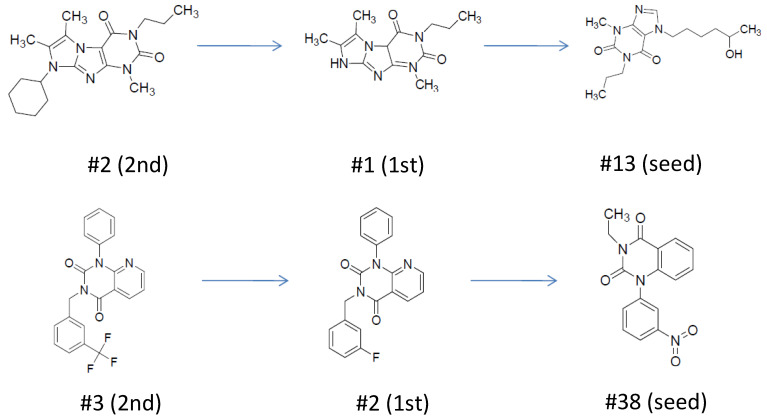
Scaffold hopping and preservation during the 2-step screening cascade in the case of PDE4B inhibitors.

**Figure 8 molecules-26-05593-f008:**
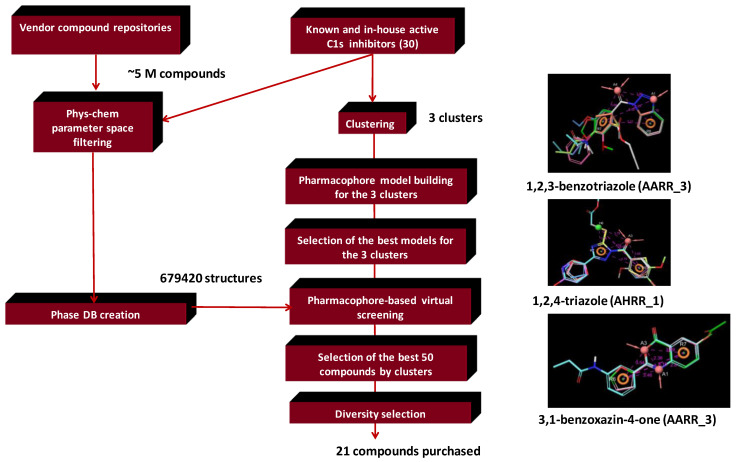
Pharmacophore model generation and application in sequential integration of virtual screening methods for identifying C1s inhibitors.

**Figure 9 molecules-26-05593-f009:**
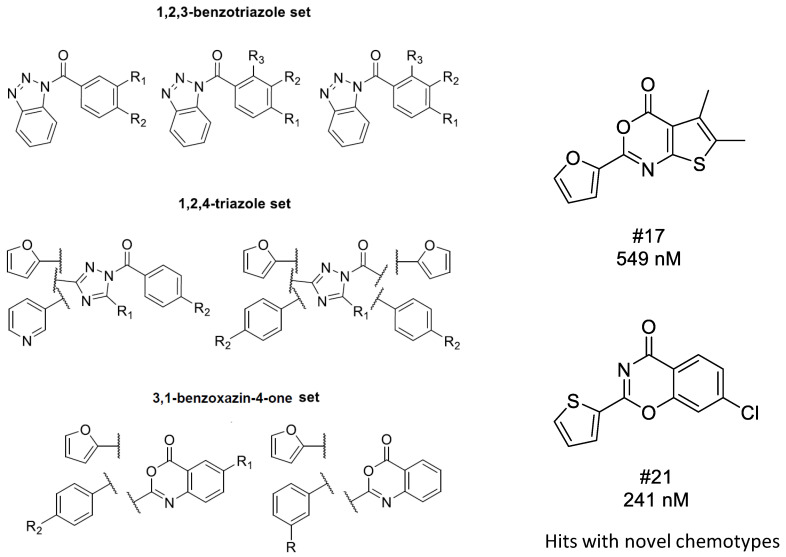
The various clusters used in the pharmacophore model building and the resulting novel chemotypes of C1s inhibitors.

**Figure 10 molecules-26-05593-f010:**
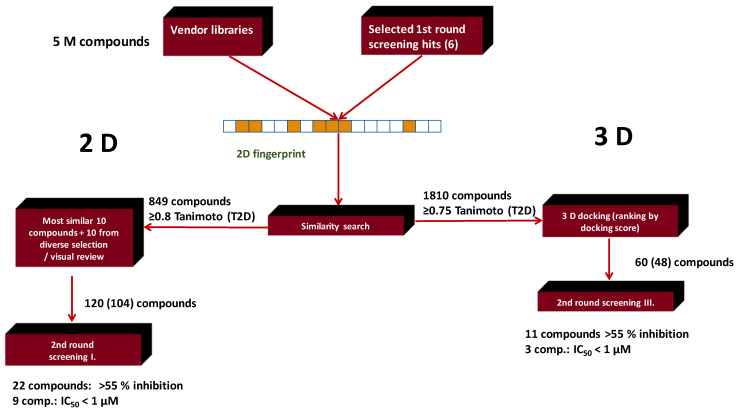
Parallel integration workflow for identification of PDE5 inhibitors.

**Figure 11 molecules-26-05593-f011:**
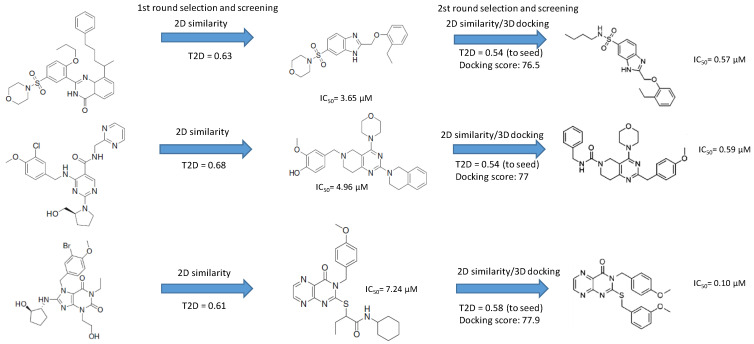
The three active PDE5 inhibitory compounds coming from the 3D docking of the 2D similarity search in the second round selection.

**Figure 12 molecules-26-05593-f012:**
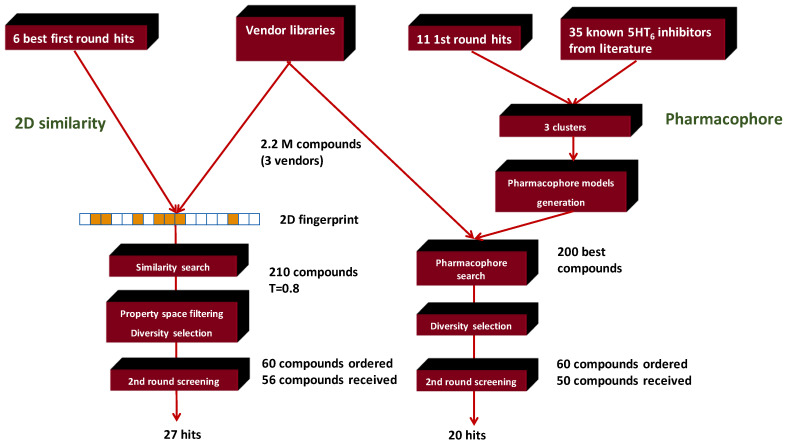
Parallel combined 2D similarity/pharmacophore matching workflow for 5 HT_6_ antagonists.

**Figure 13 molecules-26-05593-f013:**
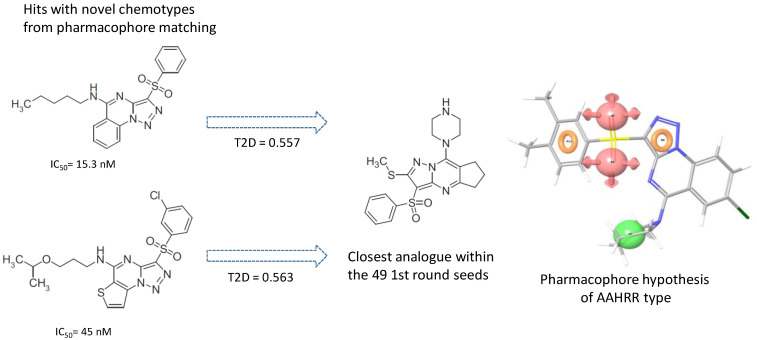
Pharmacophore model leading to novel chemotypes of 5 HT_6_ antagonists.

**Figure 14 molecules-26-05593-f014:**
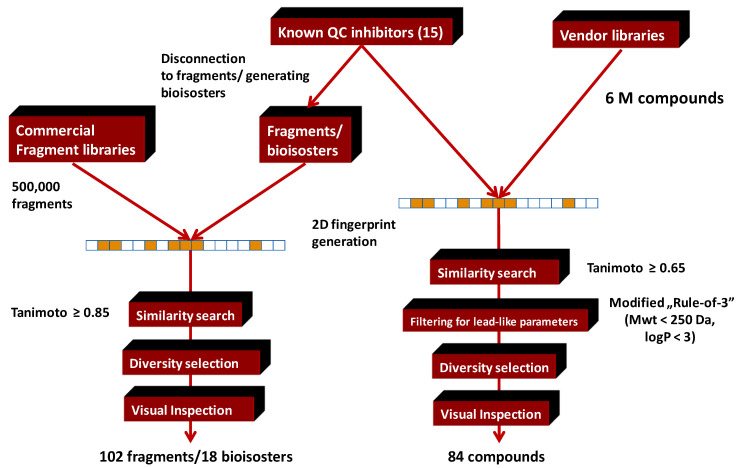
Fragment-based approaches combined with 2D similarity search.

**Figure 15 molecules-26-05593-f015:**
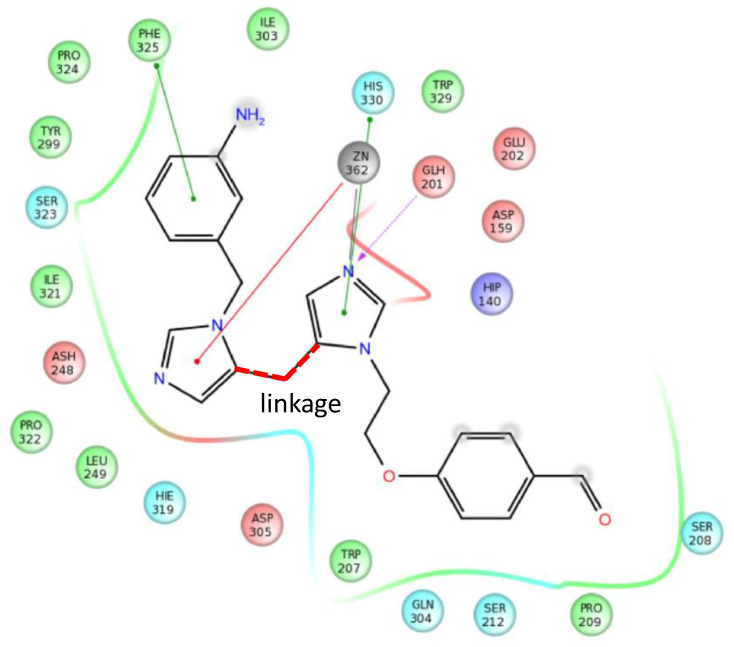
3D model of the fragment linked to glutaminyl cyclase.

**Figure 16 molecules-26-05593-f016:**
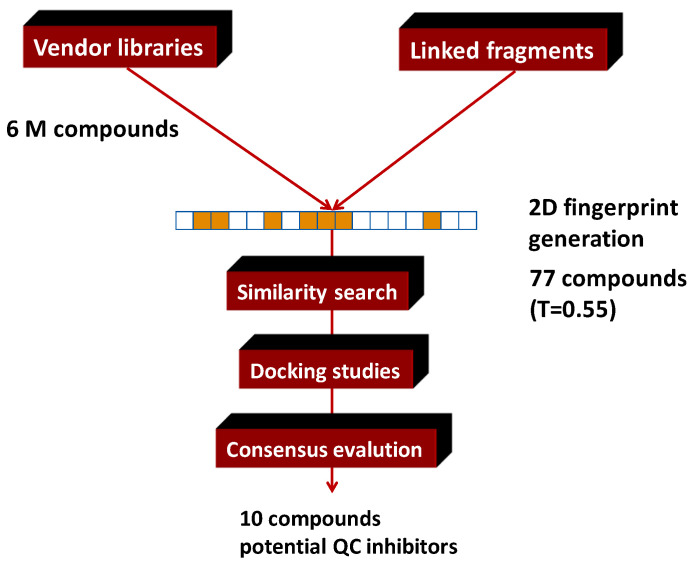
A 2D similarity selection of a focused library around the glutaminyl cyclase linked fragment.

**Figure 17 molecules-26-05593-f017:**
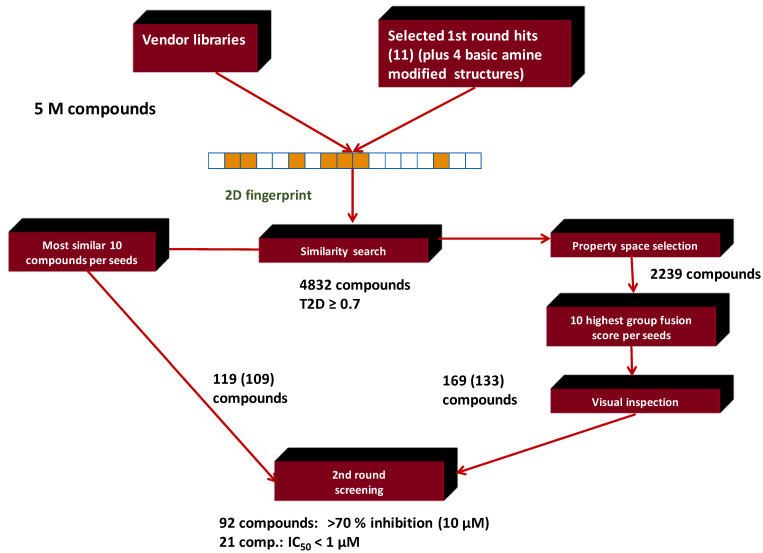
Combination of 2D similarity selection with group fusion; MCHR1.

**Figure 18 molecules-26-05593-f018:**
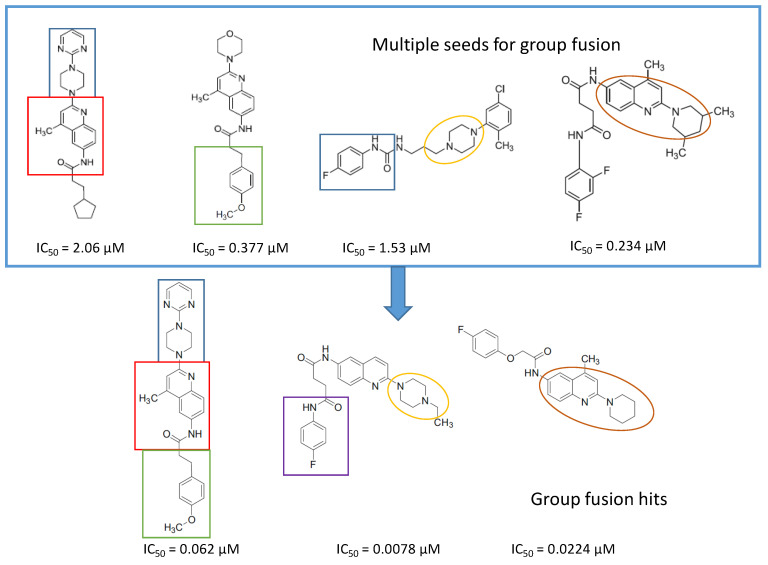
Three group fusion hits mixing the major structural motifs.

**Figure 19 molecules-26-05593-f019:**
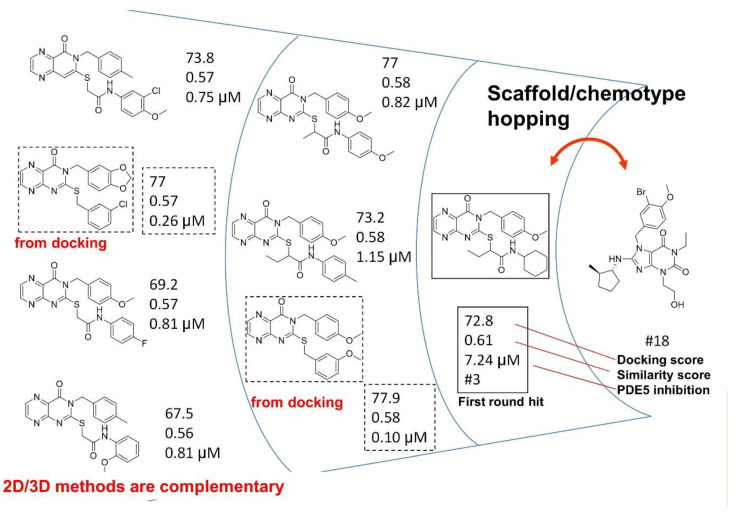
Structural evolution of PDE5 inhibitors coming from two rounds of similarity search.

**Table 1 molecules-26-05593-t001:** Comparison of the virtual screening methods (modified based on [5]).

VS Method	Input Data	Computation Demand	Hit Diversity/Novelty
**3D Docking**	Protein crystal structure (preferably with bound ligand)	↑↑	↑↑
**Pharmacophore modelling**	Several known ligands or protein crystal structure with bound ligand	↑	↑↑
**2D/3D Ligand-based similarity search**	One or several known ligands	↓	↑

**Table 2 molecules-26-05593-t002:** Combined ligand and structure-based methods (adapted from [10], with permission).

Approach	Examples	Comments
Sequential	Hierarchical VS: pharmacophore screening, application of property filters (druglikeness, ADMET), docking, manual selection	Computationally expensive methods are used at the end, on a small number of compounds Input of human expertise possible Most common strategy Has been successful for many different targets
Parallel	Parallel application of pharmacophores, similarity methods, docking, followed by automated selection	Careful selection of independent methods necessary Fully automated setup and scoring possible Promising benchmarking results
Hybrid	Protein-ligand pharmacophores, docking with pharmacophore constraints	Integration of ligand- and structure-based concept in one method Can be combined with other methods in sequential or parallel fashion

**Table 3 molecules-26-05593-t003:** Various integrated VS approaches from the recent publications.

Approaches	Databases Used for Studies	Target Proteins	Number of Hits Identified	Activities Types and Ranges	References
**Sequential integration approaches**					
Sequential combination of 2D similarity search, pharmacophore, and molecular docking	Zinc-Specs Database (441,574 compounds)	Human vascular endothelial growth factor receptor-2 (VEGFR-2)	2 hits	Inhibitory effects on the proliferation of cancer cells (U87 and MCF-7) expressing VEGFR-2.	Ai et al. [19]
Sequential approach, including 2D pharmacophore-based and structural fingerprints, ADME/Tox filtering and flexible docking	Commercial and academic libraries (58 reference ligands)	5-HT_6_R (serotonin 5-HT_6_ receptor)	6 hits	Competition binding for human serotonin 5-HT_6_R, 5-HT_2a_R, 5-HT_1a_R, 5-HT_7_R and dopaminergic D_2_R	Staron et al. [20]
Integrated in silico screening sequence (2D similarity followed by docking)	CHEMBL database, ZINC database, FDA approved drugs; molecules under clinical trials (5M compounds)	SARS-CoV-2 main protease	4 potential inhibitors	Inhibitory effect against SARS-CoV-2 main protease	Pant et al. [21]
Reverse virtual screening method: (A) VS. for identify novel scaffolds (B) 2D similarity search for hit expansion	SPECS; PKU-CNCL (Peking University)	Glycogen synthase kinase-3b (GSK-3b)	14 hits (IC_50_ = 0.71–18.2 μM)	Inhibitory effect against glycogen synthase kinase-3b (GSK-3b)	Dou et al. [22]
Sequential virtual screening: combination of 2D fingerprint matching and 3D shape modelling	CHEMBL database, FDA approved drugs	Human ether-a-go-go-related gene (hERG)	high recovery rate, maximum sensitivity, and specificity in hERG inhibition prediction	Blocking effect on human ether-a-go-go-related gene (hERG)	Schyman et al. [23]
**Parallel integration approaches**					
Parallel screening with ligand- and structure-based and virtual screening approaches	6.6 M commercial compounds for LB 1.3 M compound for docking	Leucine rich repeat kinase-2 (LRRK2)	35 compounds with IC_50_ < 10 μM	Inhibitory effect on leucine rich repeat kinase-2 (LRRK2)	Gancia et al. [24]
Parallel 2D similarity and pharmacophore-based virtual screening	SPECS	Fibroblast growth factor receptor 1 (FGFR1)	19 compounds with activity >50% at 50 μM	Inhibitory effect on fibroblast growth factor receptor 1 (FGFR1)	Gagic et a.l. [25]
Parallel 2D/3D similarity search	Pubchem	Diacylglycerol kinase alpha (DGKalpha)	17 active compounds with activity >25% at 10 μM	Inhibitory effect on Diacylglycerol kinase alpha (DGKalpha)	Velnati et al. [26]
Parallel hierarchical protocol combining ligand-based (2D similarity/pharmacophore model) and SB approaches for VS	DUD-E	Protein tyrosine phosphatase 1B (PTP1B)	10 compounds with IC_50_ at micromolar level	Inhibitory effect on protein tyrosine phosphatase 1B (PTP1B)	Yan et al. [27]
**Hybrid integration approaches**					
Hybrid integration protocol (molecular modeling methods: molecular docking, molecular dynamics simulation)	ZINC DB	Histamine H3 receptor	3 hits within micromolar and sub micromolar K_i_ range	Histamine H3 receptor antagonism	Ghamari et al. [28]
2D similarity searching, docking and scoring	PubChem library (100 M)	Histone deacetylase (HDAC)	60 virtual hit compounds; 4 active compounds	Inhibitory effect on histone deacetylase	Divsalar et al. [29]

**Table 4 molecules-26-05593-t004:** Most relevant databases applied frequently in 2D similarity search.

The Searchable Chemical Space				
Database	Unique Compounds			References
Zinc library	35 million			[43,44]
Chemspider	58 million			[45]
eMolecules	5.9 million			[46]
**The Biologically Relevant Chemical Space**				
**Database**	**Compounds**	**Bioactivity or Binding Data**	**Protein Targets**	**References**
PubChem	91,371,681	232,760,104	n.a.	[47]
ChEMBL	2,036,512	14,371,197	11.224	[48]
Binding_DB	590.985	1,328,228	6.922	[49]

**Table 5 molecules-26-05593-t005:** The calculated T2D/T3D values of the 5 PDE5 inhibitor hit compounds.

ID	Hits	IC_50_ (uM)	T2D	T3D	Seeds	Seed ID
1	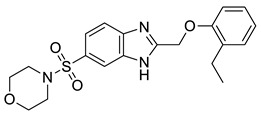	3.65	0.63	0.47	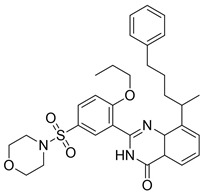	13
2	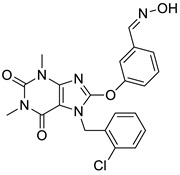	1.99	0.61	0.38	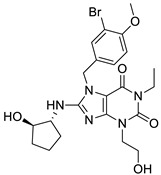	18
3	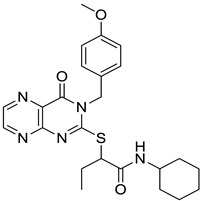	7.24	0.61	0.4	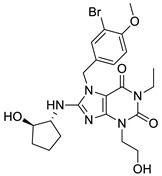	18
4	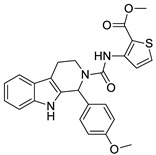	0.19	0.69	0.32	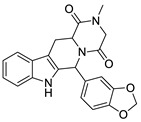	44
5	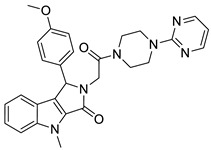	6.74	0.67	0.38	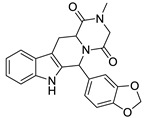	44

**Table 6 molecules-26-05593-t006:** Comparison of the 2D similarity search and the parallel integrated approach.

Search Method	Measured	Hits: Inhibition > 55% (10 μM)	Hit Rate %	Hits: IC_50_ < 1 μM	Hit Rate %
2D similarity search	104	22	21.1	9	8.6
2D similarity search plus docking	48	11	23	3	6.2

## Data Availability

Not applicable.

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
