# Peer review of "Rapid Identification of Potential Drug Candidates from Multi-Million Compounds’ Repositories. Combination of 2D Similarity Search with 3D Ligand/Structure Based Methods and In Vitro Screening"

_molecules, 2021, doi:10.3390/molecules26185593_

Round 1
Reviewer 1 Report
The manuscript entitled "Rapid identification of potential drug candidates from multi- 2 million compounds’ repositories. Combination of 2D similarity 3 search with 3D ligand/structure based methods and in vitro 4 screening" by Szilágyi et al demonstrates survey of various strategies and case studies towards virtual screening methods. Additionally, the authors have also discussed the informations from the lesson they have learnt. After carefully reading the manuscript I agree that the survey submitted can be recommended for publication with suggestions as follows.
The authors main theme of this survey focusing drugs identification from multimillion compounds repositories. In my opinion, it would be more informative, if a tabular column presented that listing current available databases of that kind.
It would be worth providing the details about contributions of available machine learning algorithms that could support the virtual screening procedures, as the contribution of ML to the science cannot be avoided.
Reviewer 2 Report
The manuscript - molecules-1356624-peer-review-v1.pdf – “Rapid identification of potential drug candidates from multimillion compounds’ repositories. Combination of 2D similarity search with 3D ligand/structure based methods and in vitro screening” is interesting from a chemistry point of view and the theme of the article meets the scope of the journal.
Considering the potential impact of the manuscript results in the research world, and with all the respect for the author's impressive work, the manuscript may be considered for publication in Molecules, after a minor version.
The manuscript has a good quality scientific, is interesting, and is worthy of publication. I'd like only to suggest some minor revisions.
- Please rewrite the abstract to highlight the results of your research.
- Please refer to the standard format of the references presented in the “Instructions for authors” of the Molecules journal and correct it accordingly. The exemplified various types of reference writing are spread throughout the manuscript (e.g.: line 239 ”… literature.xlvi,xlvii,xlviii,xlix,6” line 240 “Similarity measures.31,32…”, etc.) In the text, reference numbers should be placed in square brackets [ ] and placed before the punctuation.
- I strongly recommend to verify and rewrite some sentences to make it easy to understand. The authors are advised to clearly delimit the literature's sections from their own results. Please see the “Instructions for authors” for Research Manuscript Sections (Introduction; Results; Discussion; Materials and Methods; Conclusions; References)
- It is strongly recommended that an article present the final conclusions attesting whether or not the principal purpose has been achieved. The conclusions must also highlight the significant results obtained. A better organization of several subsections already included in the results section will be useful to easily follow the explanations and not bore the reader.
Reviewer 3 Report
In attention of the manuscript authors,
In the "molecules-1356624" manuscript, the authors have made substantial research efforts to survey and describe various combination strategies and case studies that demonstrate the efficiency of mixing the virtual screening method. Additionally, the authors discussed the lessons learned by including the advantages and pitfalls of the reported approaches. To achieve their goals, the authors developed a workflow, which mixes 2D ligand similarity-based focused libraries in the first round of in vitro biological screening, and based on the activity results, the second round hit refinement (validation) library was generated by a sophisticated (sequential or parallel) integration of various additional techniques such as 3D ligand and structure-based modeling, pharmacophore matching.
The outcomes provided by the manuscript could be a real win for scientists interested to use in silico approaches for drug discovery projects.
In this context, the potential impact of the manuscript results in the research world, and with all due respect to the author’s work, the manuscript may be considered for publication in Molecules journal following minor revision.
The main requirements addressed:
- Abstract - Please indicate more clearly the main purpose of the manuscript and the main contributions of the authors. Same recommendation for final conclusions. Also, the final sentence of the abstract need to be rewritten in a clearer way.
- Please define the terms “LB” and “SB” when first used (as the authors did for virtual screening, VS), even if these are well-known terms (section 2- first appearance in text). After these, both terms could be used abbreviated throughout the manuscript.
(e.g.line 70, “structure-based virtual screening” could be abbreviated SBVS, which is used very often in this version)
- Please pay attention to the references. Both Arabic and Roman numerals can be found in the manuscript text. (e.g. page 3, line 55, reference 47…; line 85, figure caption, adapted from Ref. 9….., line 240 “Similarity measures.31,32…, etc). Please change accordingly.
- In referee's opinion, the information contained in sections 2.1, 2.2, and 2.3 could easily be presented in a simple table (starting from table 2) along with a short description if they consider it.
Table suggestion: 1.approaches – 2.databases used for studies- 3.target proteins-4.number of hits identified-5.activities types and ranges or any other table version desired by the authors.
- The first sentence (“We developed a workflow, ….. “) of section 3., please split this sentence, it is too long and difficult to understand. These types of sentences can also be found throughout the manuscript. I strongly recommend to check the manuscript and rewrite it accordingly.
- The manuscript is too long and the reader is tempted to read superficially and possibly to mislead meaningful information. I recommend, if possible, reducing the manuscript size by keeping only the significant information to fulfill the main purpose.
